# WEIGHT-AWARE META AUXILIARY LEARNING

## ABSTRACT

Auxiliary Learning (AL) is a form of Multi-Task Learning in which a model leverages auxiliary tasks to improve performance on a primary task. AL has boosted performance across multiple domains, including navigation, image classification, and natural language processing. One of the main weaknesses of AL is the need for labeled auxiliary tasks, which can require human effort and domain expertise to generate. Furthermore, it has been shown that not all auxiliary tasks are equally beneficial to aid primary task performance. Therefore, deciding how to weight an auxiliary task or sample during training is also a hard problem. Recent work addresses the task-creation problem by learning auxiliary labels using Meta Learning approaches, often via bi-level optimization. However, these methods assume uniform weighting across data points. Other works present selecting weights for known tasks. In this work, we propose Weight-Aware Meta Auxiliary Learning (WAMAL), a novel framework that jointly learns both auxiliary labels and per-sample auxiliary loss weights to better guide the main task. Our method improves upon existing approaches by allowing more nuanced and adaptive task supervision. Across multiple benchmarks WAMAL surpasses both handcrafted auxiliaries and prior meta-auxiliary baselines. On CIFAR-100 (20 super-classes, VGG16) it reaches 80.2% test accuracy (+5.6 pp over human-designed auxiliaries; +2.8 pp over weight-unaware meta-learning). When fine-tuning ViT-B/16 on Oxford-IIIT Pet, WAMAL improves accuracy by 0.62 pp. These results underscore the importance of learning both which auxiliary tasks to use and how strongly to weight them at the sample level. Code repo will be released after submission. Anonymized version: `https://anonymous.4open.science/r/wamal-66EF/README.md`.

## 1 INTRODUCTION

**Motivation.** Auxiliary Learning (AL) improves generalization by utilizing additional supervision, it is limited by two considerations: *(i)* which auxiliary labels to use, and *(ii)* how strongly each training sample should influence the auxiliary loss. Handcrafted labels may be suboptimal or unavailable, and uniform weighting can emphasize the wrong information in regard to helping the primary task. We aim to address both choices jointly, within a bi-level framework that directly optimizes for primary-task improvements.

Auxiliary Learning (AL) is a technique by which learned or pre-labeled auxiliary tasks are provided as an additional objective to a network during its training with the intended goal of improving the network's performance on a desired primary task. Auxiliary Learning can be thought of as a sub-field of multi-task learning, in which the objective of the training is to improve the main network's performance on the primary task while the auxiliary tasks regularize the training (Caruana, 1997; Ponti, 2021). It has been demonstrated that the inclusion of auxiliary tasks during training improves generalization and network performance on unseen samples across a large range of domains, including speech recognition, navigation, and image classification (Jaderberg et al., 2016; Goyal et al., 2019; Mirowski et al., 2016; Liebel and Körner, 2018; Toshniwal et al., 2017). Even small, tangentially related tasks have been shown to provide significant support to the main task (Liebel and Körner, 2018). The intuition is that using the auxiliary task pushes the network to learn a shared representation of the data that guards against overfitting on the primary task (Liu et al., 2019).

A historic weakness of Auxiliary Learning has been the need for additional human labeling during the creation of supervised auxiliary tasks. This requires a large amount of human effort and domain expertise for each auxiliary task. Furthermore, we know that poor task selection can ultimately harm

primary task performance (Gururangan et al., 2020). Therefore, the manner in which primary and auxiliary tasks should be optimally combined during the weight update procedure can be ambiguous and require expert knowledge.

Meta Auxiliary Learning (MAXL) attempts to alleviate the problem of auxiliary task labeling through the procedural generation of an auxiliary task that optimizes the performance on the given primary task (Liu et al., 2019). The MAXL framework works by organizing the inputs of the primary task into hierarchical subclasses for each primary class using an additional label generation network. As such, MAXL is one of several approaches for dynamic label generation that utilize bi-level optimization (Navon et al., 2020; Chen et al., 2023). Bi-level optimization, which is at the heart of many Meta Learning procedures, arises here as the gradients of the label network are calculated with respect to the performance of the main network on the primary task, resulting in a Hessian-inverse vector calculation and generally increased implementation complexity.

In this work, we will extend the bi-level optimization approaches for Meta Auxiliary Learning by learning sample-level auxiliary weights as the auxiliary task is learned. Our approach involves adding a weight-selection head to a label generation network that enables the new network to select a new auxiliary task label and decide how much that sample should be weighted during the auxiliary training. Several techniques have been explored in the literature to optimize auxiliary task weighting but our novel focus is the dynamic selection of the auxiliary task labels and also the weights at a sample-level (Kung et al., 2021; Grégoire et al., 2023; Abbas and Tap, 2019; Chennupati et al., 2019). We demonstrate that WAMAL outperforms single-task learning and weight-unaware Meta Auxiliary Learning on a variety of tasks and network architectures.

Enabling a network architecture to work with a bi-level optimization-based auxiliary task generation often requires bespoke implementations for each chosen architecture. To help alleviate the burden of adapting new architectures, our framework presents generic wrappers that facilitate the conversion of virtually any image classification network into WAMAL-ready (and MAXL-ready) primary networks and label-weight generating networks. This will be expanded to more generic use-cases in the future. We demonstrate that this approach works across several architectures and has enabled us to be perhaps the first to use Vision Transformers (ViTs) in the Meta Auxiliary Learning space (Dosovitskiy et al., 2020).

**Contributions.** (1) We introduce **WAMAL**, a bi-level framework that jointly learns auxiliary labels and per-sample auxiliary weights to guide the primary objective. (2) We motivate bounded weights as a mechanism to stabilize the bi-level optimization and explore the role of the entropy term in preventing label collapse. (3) We provide generic Python wrappers that turn standard image classifiers into WAMAL-ready primary and label-weight networks, lowering the barrier to adoption. (4) Across VGG16, ResNet50, and ViT-B/16, WAMAL improves over single-task training and weight-unaware meta-auxiliary baselines.

In summary, we demonstrate a method to dynamically learn sample-level loss weights and an auxiliary task at the same time using bi-level optimization. Our approach, WAMAL, significantly improves main network performance on a variety of tasks and provides greater improvement than weight-unaware approaches. This paper focuses on image classification. However, we discuss how the framework extends to other domains, leaving experimental verification to future work.

## 2 RELATED WORK

### 2.1 AUXILIARY AND MULTI-TASK LEARNING IN VISION

Multi-task Learning (MTL) is a well-studied and widely used method to have a network learn multiple tasks simultaneously (Caruana, 1997; Liebel and Körner, 2018; Zhang and Yang, 2017). Auxiliary Learning (AL) is a special case of MTL, in which there is one primary task of importance and one or more auxiliary tasks that support the performance of the main task (Liu et al., 2019). MTL and AL have been shown to improve target task performance compared to networks trained on a single task, particularly in low data contexts (Standley et al., 2019; Zhang and Yang, 2017). There has been extensive work using MTL and AL in the vision domain, such as the auxiliary classifiers in GoogLeNet (Szegedy et al., 2014), multi-task cascaded convolutional networks (Zhang et al., 2016), and state-of-the-art performance on three vision tasks using one convolutional network (Eigen and

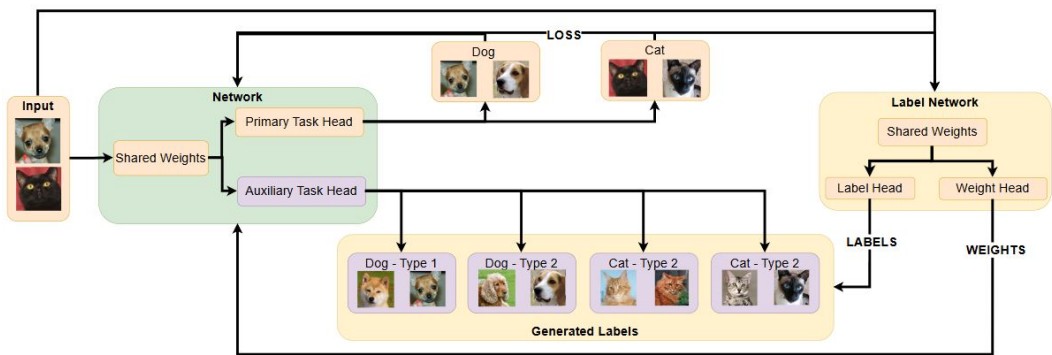

Figure 1: Sample WAMAL Setup

Fergus, 2014), among many others (Strezoski et al., 2023; Kokkinos, 2017; Rasmus et al., 2015). Figure 4 shows a classic auxiliary task setup and a label-network auxiliary task setup.

## 2.2 TASK WEIGHTING

The standard loss formulation in Auxiliary Learning is given by:

$$\mathcal{L}_{total}(\theta) = \mathcal{L}_{prim}(\theta) + \sum_{i=1}^{k} \lambda_i \mathcal{L}_{aux}^{(i)}(\theta) \tag{1}$$

Where $k$ is the number of auxiliary tasks, $\lambda_i$ represents the weight of the task $i$, and the network is parametrized by $\theta$. The Taskonomy framework provides a methodology to investigate the relationship between candidate auxiliary tasks and map their relationship based on their contribution to target task performance (Zamir et al., 2018). (Standley et al., 2019) took this a step further and demonstrated that not all possible tasks are helpful for learning. As such, several papers in the field have attempted to learn optimal $\lambda_i$ loss weights on a per-task basis . (Kendall et al., 2017) introduced the idea of using homoscedastic aleatoric uncertainty to weight known auxiliary tasks. (Liebel and Körner, 2018) extended this work but enforced positive regularization values to achieve improved results (Gong et al., 2019). GradNorm demonstrated strong results by normalizing gradient magnitudes on a per-task basis (Chen et al., 2017) . Task weighting as a Pareto multi-objective optimization was also attempted (Sener and Koltun, 2018).

While the previously mentioned approaches attempt to weight on a per-task basis, SLGrad presents a sample-level weighting approach for known auxiliary tasks that scales each sample on the cosine-similiarity of the sample's loss gradient and the primary task's validation gradient (Grégoire et al., 2023). Auxilearn also presents a framework to learn sample-specific weights for a known task using bi-Level optimization with implicit differentiation (Navon et al., 2020).

## 2.3 TASK GENERATION

MAXL started the label network paradigm for dynamic task generation in the Auxiliary Learning space using Bi-Level Optimization (Liu et al., 2019). The MAXL framework trains a label network to create an auxiliary task that optimizes the main network's performance on the target primary task. Figure 5 in the Appendix shows how the setup of a label network is used to generate the auxiliary task. For a label network $g^{\phi}$, the per-sample loss can be constructed as:

$$\ell_{total}(x_i, y_i) = \ell_{primary}\big(f_{primary}^{\theta}(x_i), y_i\big) + \lambda \, \ell_{aux}\big(f_{aux}^{\theta}(x_i), g^{\phi}(x_i)\big) \tag{2}$$

Bi-Level optimization approaches attempt to find the optimal primary network weights $\theta^*$ and auxiliary labeling network weights $\phi^*$ that satisfy:

$$\phi^* = \arg\min_{\phi} \; \mathcal{L}_{aux}\big(\theta^*(\phi)\big), \text{ s.t. } \theta^*(\phi) = \arg\min_{\theta} \; \mathcal{L}_{prim}(\theta, \phi) \tag{3}$$

The gradient update of the label network is given by:

$$\nabla_\phi \mathcal{L}_{aux}\big(\theta^*(\phi)\big) = -\nabla_\theta \mathcal{L}_{aux} \cdot \big(\nabla_\theta^2 \mathcal{L}_{prim}\big)^{-1} \cdot \nabla_\phi \nabla_\theta \mathcal{L}_{prim} \tag{4}$$

Auxilearn, another approach using Bi-Level Optimization, attempted to use Neumann approximation to optimize this update (Navon et al., 2020). Other Bi-Level Optimization approaches in task creation involve generating features/samples on the fly and finding useful "questions" as general value functions (Veeriah et al., 2019; Chen et al., 2023).

There have been several attempts at task generation without Bi-Level optimization. These approaches generally involve selecting task objectives from a predefined or procedurally generated pool of tasks. Approaches include using Beta-Bernoulli multi-armed bandit framing (Guo et al., 2019), search over unified taxonomy (Dery et al., 2023), and trial-and-error search over generated features (Rafiee et al., 2023).

## 3 PROBLEM FORMULATION

This work focuses on learning sample-level weights for the auxiliary task in addition to learning auxiliary task labels.

Let our dataset be represented as $D = \{(x_i, y_i)\}_{i=1}^N, x \in X, y \in Y$. We construct a label-weight selection network $g^\phi$ that maps an input in $X$ to a new set of labels $\hat{Y}$ and weight between $[2^{-r}, 2^r]$, where $r$ is a hyperparameter representing the range of the weight output. The label network will have two heads represented as $g_{label}^\phi : X \to \hat{Y}$ and $g_{weight}^\phi : X \to [2^{-r}, 2^r]$. The size of $\hat{Y}$ is an integer multiple of the size of $Y$, s.t. $\psi|Y| = |\hat{Y}|$ for some $\psi \in \mathbb{Z}$. $\psi$ represents the hierarchy factor by which each main task label is expanded into auxiliary sub-classes. We have found $r = 5$ to provide an effective range for weight selection and this selection is substantiated in our ablation experiments.

We will train a main network $f$, parameterized by $\theta$, that has two output heads $f_{prim}^\theta : X \to Y$ and $f_{aux}^\theta : X \to \hat{Y}$. We will train the network with the following per-sample loss:

$$\ell_{total}(x_i, y_i) = \ell_{prim}\big(f_{prim}^\theta(x_i), y_i\big) + g_{weight}^\phi(x_i)\, \ell_{aux}\big(f_{aux}^\theta(x_i), g_{label}^\phi(x_i)\big) \tag{5}$$

The weight update of the $g^\phi$ network is done only with respect to the primary loss:

$$\ell_{prim}(f_{prim}^\theta(x_i), y_i) \tag{6}$$

And the gradient update is still calculated as in Equation 4. Figure 1 provides a visual representation of WAMAL's setup.

### 3.1 LABEL HIERARCHY AND LOSS ENTROPY

MAXL employs hierarchy-constrained label generation through the use of a Masked Soft-max (Liu et al., 2019). In other words, each class in the primary task will have some fixed number of subclasses in the auxiliary task.

If $z$ are the logits generated from an input $(x_i, y_i)$ and $y_i$ represents the integer index of the primary label, then the following gives the probability for auxiliary label $k$:

$$p_k = \frac{\exp(z_k)m_k}{\sum_{j=1}^K \exp(z_j)m_j}, m_k = \begin{cases} 1 \text{ if } \psi \cdot y_i \le k < \psi \cdot (y_i + 1) \\ \text{else } 0 \end{cases} \tag{7}$$

where $K$ is the number of auxiliary classes, $z$ represents raw logits, and $m \in \{0, 1\}^K$ acts as the mask. MAXL employs a hierarchy factor $\psi$ that dictates how many subclasses a primary task class will have in the auxiliary task. Masking auxiliary logits with a per-class partition (Eq 7) guarantees coverage of each primary class with a fixed number of subclasses. This minimizes complexity and

provides a straightforward guard against collapse. Alternative approaches are possible but are out of the scope of this work.

In order to avoid collapsed auxiliary class labels, MAXL uses an entropy factor over the label space that is added to the loss. This encourages the label network to use the entire auxiliary class space. We can formulate this entropy factor in our setup as $\mathcal{H}(g_{label}^{\phi}(x)_{(b)})$ per evaluation batch $b$:

$$\mathcal{H}(g_{label}^{\phi}(x)_{(b)}) = \sum_{k=1}^{K} g_{label}^{\phi}(x)_{(b)}^{k} \, \log g_{label}^{\phi}(x)_{(b)}^{k}, \quad g_{label}^{\phi}(x)_{(b)}^{k} = \frac{1}{N} \sum_{n=1}^{N} g_{label}^{\phi}(x)_{(b)}^{k}[n] \quad (8)$$

where $K$ is again the number of auxiliary classes and N is the batch size. We ran ablations over the impact of the weight of this entropy factor as part of our experimentation.

Focal loss is used after the Masked Soft-max to promote the use of the entire auxiliary label space.

## 4 METHODOLOGY

### 4.1 NETWORKS

We experiment with various network architectures. The majority of our experimental setups have the backbone architecture of the main network match the backbone of the label-weight generation network. In addition to this, we conducted ablations in which the two backbones are network architectures of different sizes/capacities. We leave the exhaustive exploration of the optimization of the size of the two backbone networks for future work. The results presented in this paper focus on the performance of the WAMAL framework using VGG16 (Liu and Deng, 2015), ResNet50 (He et al., 2015), and Vision Transformer (ViT-B/16) (Dosovitskiy et al., 2020) as the backbone.

#### 4.1.1 WEIGHT SELECTION

Our label-weight generation network is fitted with a weight-head composed of 2 linear layers that can output a scalar auxiliary sample weight. The weight for sample $x_u$ is generated as follows:

$$w_u = \sigma(W_2 \cdot \textbf{ReLU}(W_1 g_{backbone}^{\phi}(x_u) + b_1) + b_2) \quad (9)$$

The selected unscaled weight $w_u$, is then used to get the true selected weight as:

$$2^{2rw_u - r} \quad (10)$$

This gives us constrained auxiliary sample weights $w \in [2^{-r}, 2^r]$ as in Equation equation 10. Bounded weights (i) prevent dominance or vanishing of the auxiliary term, (ii) promote numerically stable implicit-differentiation steps as it constrains the norm of Hessian–vector product. A theoretical justification can be found in the appendix. Our ablations over this hyperparameter reveal the importance of constraining these weights and justify our selection of $r$ value.

#### 4.1.2 WAMAL/MAXL WRAPPER

To enable rapid experimentation and flexible integration of novel architectures into the WAMAL framework, we introduce two lightweight Python modules that can generically wrap any standard image classification backbone. These wrappers create a level of abstraction that allows an arbitrary network to be deployed in a Meta Auxiliary Learning pipeline with low overhead. Both work by first stripping the backbone network of its final classification layer.

The primary network wrapper then adds primary and auxiliary task heads, which are two-layer feedforward networks. The interface exposes parameter overrides to enable gradient-based Meta Learning approaches that require inner-loop optimization.

The label-weight generation network adds a classifier head, to attribute auxiliary task labels, and a scalar weight head, which outputs a value in the range of $[0, 1]$ as defined in Equation 9. This framework is suitable both for training networks from scratch and fine-tuning pretrained architectures.

**Remark (entropy against collapse).** In the MAXL framework, it was shown that adding batch entropy on $g_{\text{label}}^{\phi}$ discourages degenerate solutions where only a few subclasses are used. We study the impact of this entropy on our framework in our ablations.

## 5 EXPERIMENTAL SETUP

This section summarizes datasets, preprocessing, model configurations, and training schedules used across experiments.

### 5.1 DATASETS

CIFAR-10 and CIFAR-100 have become ubiquitous benchmark datasets in the Auxiliary Learning space(Grégoire et al., 2023; Chen et al., 2023; Fan et al., 2018; Cubuk et al., 2018; Liu et al., 2019; Navon et al., 2020; Krizhevsky, 2009).

CIFAR-100 provides a 20 superclass hierarchy. Each of the 100 classes in the dataset is mapped into one of the 20 Superclasses such that each superclass is composed of 5 of the 100 classes (Krizhevsky, 2009). Table 2 in the Appendix shows how the Superclasses are constructed. This is a very useful human-labeled auxiliary task benchmark, against which we can compare our proposed methodology.

We additionally evaluate on SVHN (Street View House Numbers) (Netzer et al., 2011), Oxford–IIIT Pet (Parkhi et al., 2012), Food101 (Bossard et al., 2014), and CUB-200-2011 (Wah et al., 2011)

### 5.2 TRAINING

We adopt a standard alternating training procedure in which the label-weight generation network is trained for one epoch, followed by one epoch of main network training on both the primary and auxiliary tasks. Our training algorithm is outlined in Section A.13 in the Appendix. In all experiments we capture performance metrics after one round of training for both networks.

All experiments, unless otherwise noted, were trained using Stochastic Gradient Descent (for both the primary and label-weight networks), Hierarchy Factor($\psi$)=5, Range Hyperparameter($r$)=5, Label-Weight Network Optimizer Decay=5e-4, Scheduler Step Size=50, and Scheduler Gamma($\gamma$)=0.5. The default training parameters for the VGG16 architecture were: Primary Learning Rate=1e-2, Label-Weight Learning Rate=1e-3. The default training parameters for ViT-B/16 were: Primary Learning Rate=5e-4, Label-Weight Learning Rate=1e-3. For ResNet50, the default parameters were: Primary Learning Rate=1e-2, Label-Weight Learning Rate=1e-2.

When training on the CIFAR and SVHN datasets, the provided 32x32 resolution images were used. All other datasets were resized to 224x224. Standard transformations were applied for each dataset. When fine-tuning the ViT architecture, the provided image processor was applied to the images during training and testing.

All experiments that involved training networks from scratch were trained for 200 epochs. All fine-tuning experiments were trained for 75 epochs.

## 6 RESULTS AND DISCUSSION

We present our results comparing WAMAL to standard Single-Task Learning (STL) and Meta Auxiliary Learning (MAXL) task generation.

### 6.1 EXPERIMENTS

#### 6.1.1 VGG16

As mentioned previously, the CIFAR100 dataset provides a 20-Superclass hierarchy. We conducted an experiment to compare a VGG16 backbone architecture trained from scratch on the 20-Superclass CIFAR100 task. In addition to STL, MAXL and WAMAL, we will also use the 100-class labels as a human auxiliary task (as was done in (Liu et al., 2019)).

Table 1: Comparison of test accuracy across different architectures and tasks using Single Task, MAXL, and WAMAL (in percentage), with SGD optimizer used throughout. Results reflect the average accuracy of 3 runs.

| Task | Architecture | Optimizer | Accuracy (%) | | |
|------|-------------|-----------|-------------|------|-------|
| | | | Single Task | MAXL | WAMAL |
| SVHN | VGG16 | SGD | 0.9395 ± 0.0003 | 0.9454 ± 0.0012 | **0.9554 ± 0.0012** |
| CIFAR100-20 | VGG16 | SGD | 0.7378 ± 0.0028 | 0.7726 ± 0.0005 | **0.8022 ± 0.0023** |
| CIFAR10 | ResNet50 (fine-tune) | SGD | 0.8783 ± 0.0015 | 0.9035 ± 0.0021 | **0.9114 ± 0.0012** |
| CUB200 | ResNet50 (fine-tune) | SGD | 0.6981 ± 0.0046 | 0.7110 ± 0.0039 | **0.7179 ± 0.0036** |
| CUB200 | ViT-B/16 (fine-tune) | SGD | 0.8132 ± 0.0011 | 0.8113 ± 0.0009 | **0.8136 ± 0.0012** |
| Oxford-IIIT Pet | ViT-B/16 (fine-tune) | SGD | 0.9222 ± 0.0028 | 0.9221 ± 0.0013 | **0.9284 ± 0.0007** |
| Oxford-IIIT Pet (30%) | ViT-B/16 (fine-tune) | SGD | 0.9190 ± 0.0045 | 0.9160 ± 0.0021 | **0.9225 ± 0.0016** |
| Food101 | ViT-B/16 (fine-tune) | SGD | 0.8665 ± 0.0006 | 0.8587 ± 0.0008 | **0.8787 ± 0.0010** |

As is clear from Figure 2a, our WAMAL (80.22%) approach significantly outperforms both MAXL (77.26 %) and the human auxiliary task (74.64%).

### 6.1.2   ViT-B/16

We ran several fine-tuning experiments using a pretrained ViT-B/16 backbone. We ran full fine-tuning experiments on CUB200, Oxford-IIIT, and Food101. WAMAL did notably well on the Food101 task, providing a 1.2% performance improvement over both approaches. The Oxford-IIIT Pet task also saw a notable 0.62% improvement with the assistance of WAMAL. We also see a modest improvement for the CUB200 dataset, which may be attributed to the large amount of sample data in the training set.

We also conducted additional fine-tuning experiments using only 30% of the Oxford-IIIT Pet dataset. In this data-constrained setup, we notice an improvement with our approach.

### 6.1.3   Range Ablation

We study the impact of $r$ selection by training VGG16 on the CIFAR-100 20 Superclass problem with $r$ selections in the range [0,40]. We see significant training instability at and above $r = 10$, with $r = 40$ preventing convergence entirely. We also see that low values ($r = 1.25$, 79.8%) perform worse than larger viable values. This further emphasizes the significance of sample weighting overall. A selection in the range [2.5,5] is optimal (r=5, 81.6%). This supports the claim that weight constraining is critical to training stabilization in this framework.

### 6.1.4   Adam Ablation

We ran ablation experiments to observe the impact of using Adam as the optimizer in place of SGD with WAMAL. We trained VGG16-based WAMAL architecture using a wide range of learning rates. Our results, as seen in the Technical Appendix, reveal that using Adam makes the training setup sensitive to learning rate selection. Adam (79.3%) does not perform as well as SGD (80.9%) even with an exhaustive search for learning rate. Adam's adaptive learning rate and momentum may compound with large auxiliary task weights, resulting in suboptimal training curves.

### 6.1.5   Asymmetric Backbone Capacities Ablation

We study the effect of using backbones with different capacities in the primary and label backbones using the CIFAR-10 dataset. We see that using a smaller backbone network can provide results on par with using the same primary network backbone. Using a larger label network than the primary network does not provide additional benefit. We leave an exhaustive exploration into the trade-offs of different model capacities to future work.

### 6.2   Weight Analysis

To understand the contribution of the weight selection on the primary task training, we explore how the label-weight network learns to attribute weights.

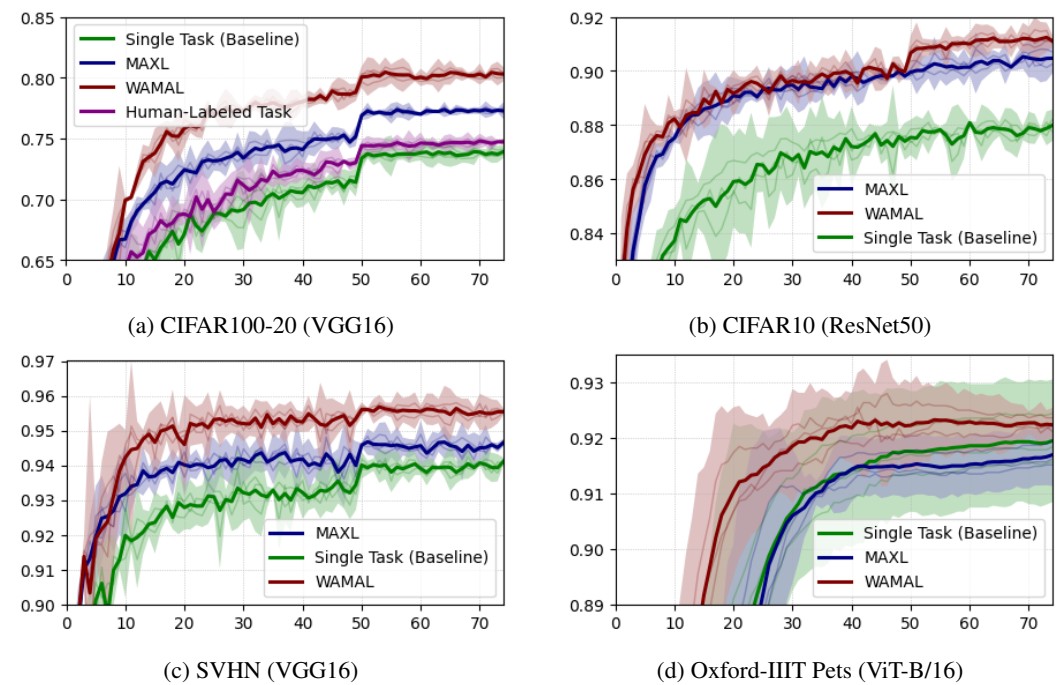

Figure 2: Sample test accuracy curves over multiple runs for WAMAL, MAXL, and Single Task training setups across datasets. Additional curves can be found in the appendix.

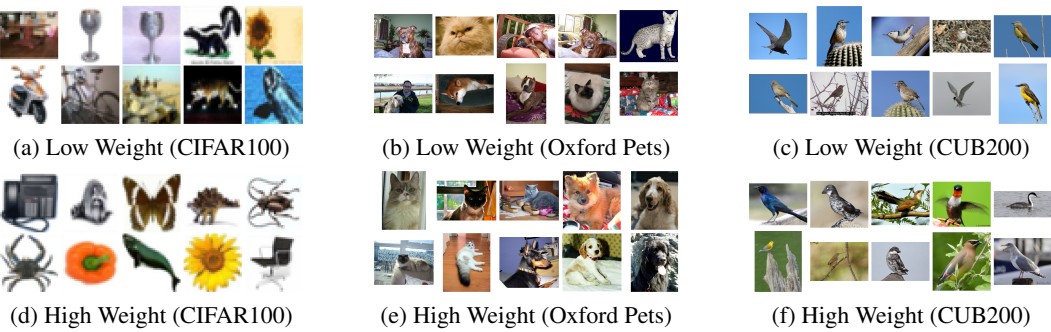

(a) Low Weight (CIFAR100)  (b) Low Weight (Oxford Pets)  (c) Low Weight (CUB200)

(d) High Weight (CIFAR100)  (e) High Weight (Oxford Pets)  (f) High Weight (CUB200)

Figure 3: High and Low Weight Training Samples.

### 6.2.1 WEIGHT DISTRIBUTION ANALYSIS

We collected the distribution of the unscaled weights (prior to the application of Equation 10) attributed to training datapoints from several model and task setups. We present the results to show how our label-weight generation network is attributing weights to the training data.

As we can observe in Figure 11 in the Appendix, the label-weight generation networks tend to learn a normal-like distribution across all training data-points. The label-weight networks also clearly use a wide range of values within their selection space and do not seem to be limited by our choice of $r$ value. It is interesting to note that the different setups have different weight biases. This supports the idea that our weight selector learns how well the primary task and network respond to auxiliary tasks during training.

WAMAL yields larger gains when auxiliary structure is rich (e.g., many fine-grained modes) and the base model has headroom, while improvements diminish as base accuracy saturates (e.g., strong ViT fine-tuning). This suggests that learned sample-level weighting is most impactful when the auxiliary signal is heterogeneous and not uniformly useful.

### 6.2.2 QUALITATIVE ANALYSIS

We collected and visually examined the images with the highest and lowest weights from the label-weight generation model.

Qualitatively, there seems to be a trend that the lower-weighted samples are generally of lower quality and clarity. The low-weight CIFAR100 samples, for example, seem to be noisier and less representational than their high-weight counterparts. The high-weight samples in CIFAR100, generally contain distinct, recognizable objects. Similarly, the low-weight Oxford-IIIT samples include images that contain humans and less informational perspectives. The high-weight CUB200 images tend to have more colorful and novel birds and varied backgrounds compared to the low-weight images.

### 6.2.3 ETHICAL CONSIDERATIONS

Per-sample weighting can unintentionally underweight minority classes or lead to the discarding of safety-related samples. We recommend monitoring per-class weight statistics as it pertains to safety.

### 6.3 LIMITATIONS

Although WAMAL yields strong results across various tasks and architectures, it exhibits some limitations in specific scenarios:

**Performance on ViT with Abundant Data -** WAMAL and MAXL did not provide significant gains over Single-Task learning when fine-tuning ViT-B/16 on CUB200. One potential reason is that ViT's can learn robust representations in the presence of an abundance of data, reducing the benefit of auxiliary tasks.

**Instabilities with Adam -** We observed that training WAMAL with Adam could lead to less stable learning. Adam's adaptive learning rate and momentum may compound with large auxiliary task weights, resulting in suboptimal training curves.

## 7 CONCLUSION

In this work, we presented WAMAL, a novel bi-level optimization framework that jointly learns auxiliary labels and per-sample weights to enhance a primary classification task. Our experiments show that incorporating sample-level weighting alongside task label generation yields performance gains over both single-task training and weight-unaware Meta Auxiliary Learning across multiple datasets and architectures. Notably, WAMAL surpasses both human-designed and MAXL-based auxiliary tasks on the CIFAR-100 20-Superclass benchmark. We also demonstrate that WAMAL improves fine-tuning results for ResNet50 and ViT-B/16, including a 1.2% accuracy boost on Food101 and a 0.62% boost on Oxford-IIIT Pet. Our analysis suggests that adaptively emphasizing certain training samples significantly strengthens auxiliary-task effectiveness.

In addition, we introduce an open-source library that converts arbitrary image classifiers into WAMAL-ready primary and label-weight generation networks. This library facilitates rapid experimentation with Meta Auxiliary Learning pipelines.

Looking ahead, we plan to extend our approach beyond image classification. Similar dynamic sample-level weighting could be explored in NLP or applied to tasks like image segmentation and dense prediction. Another promising direction is multi-task learning, where an auxiliary task could be crafted to improve multiple target tasks simultaneously instead of focusing on a single task.

Overall, these results highlight the importance of learning not only what auxiliary tasks to use, but also how to assign weights to samples dynamically. We hope this work will spur further development and refinement of Meta Auxiliary Learning frameworks.

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

# A APPENDIX

## A.1 CIFAR100 20-SUPERCLASS PROBLEM

| Superclass | Classes |
|---|---|
| Aquatic mammals | beaver, dolphin, otter, seal, whale |
| Fish | aquarium fish, flatfish, ray, shark, trout |
| Flowers | orchids, poppies, roses, sunflowers, tulips |
| Food containers | bottles, bowls, cans, cups, plates |
| Fruit and vegetables | apples, mushrooms, oranges, pears, sweet peppers |
| Household electrical devices | clock, computer keyboard, lamp, telephone, television |
| Household furniture | bed, chair, couch, table, wardrobe |
| Insects | bee, beetle, butterfly, caterpillar, cockroach |
| Large carnivores | bear, leopard, lion, tiger, wolf |
| Large man-made outdoor things | bridge, castle, house, road, skyscraper |
| Large natural outdoor scenes | cloud, forest, mountain, plain, sea |
| Large omnivores and herbivores | camel, cattle, chimpanzee, elephant, kangaroo |
| Medium-sized mammals | fox, porcupine, possum, raccoon, skunk |
| Non-insect invertebrates | crab, lobster, snail, spider, worm |
| People | baby, boy, girl, man, woman |
| Reptiles | crocodile, dinosaur, lizard, snake, turtle |
| Small mammals | hamster, mouse, rabbit, shrew, squirrel |
| Trees | maple, oak, palm, pine, willow |
| Vehicles 1 | bicycle, bus, motorcycle, pickup truck, train |
| Vehicles 2 | lawn-mower, rocket, streetcar, tank, tractor |

Table 2: CIFAR-100 20 Superclass and Corresponding Single Classes

## A.2 Auxiliary Task Setups

We provide a visual sample of the difference between the training setups for Auxiliary Learning in three scenarios: (i) The auxiliary task is known/provided by human labels, (ii) The auxiliary task is created from a label network (MAXL), (iii) The auxiliary task and per-sample weights are generated by WAMAL.

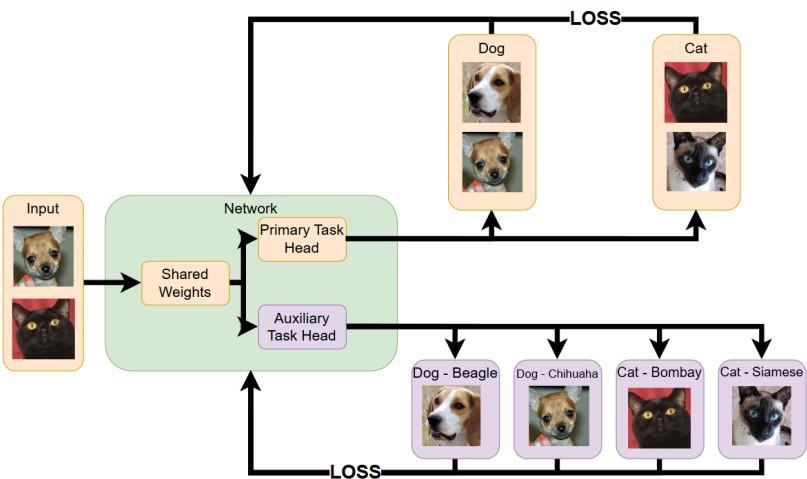

Figure 4: Classic Auxiliary Task Setup

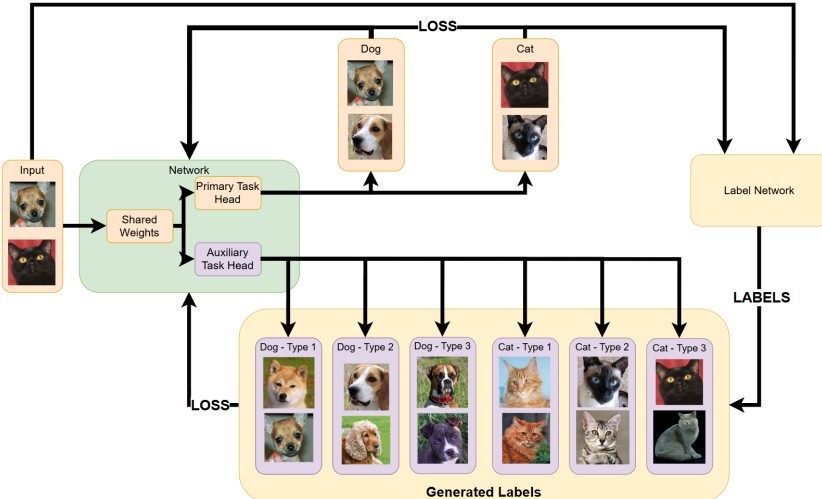

Figure 5: Label Network Training Setup

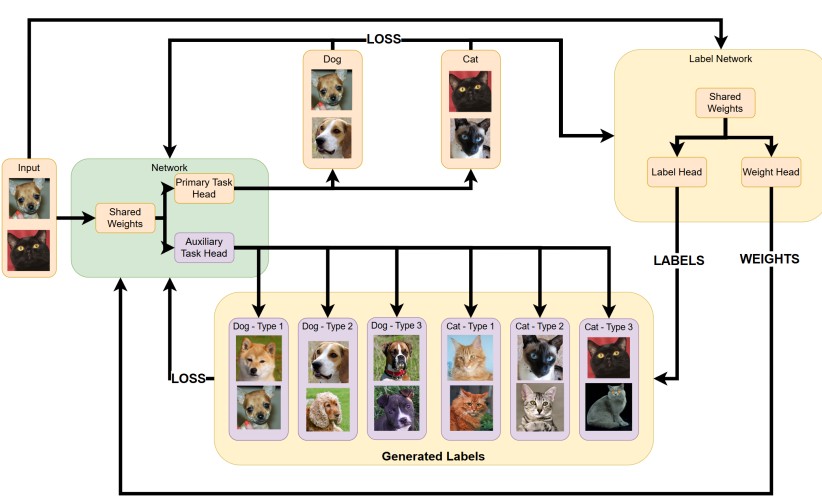

Figure 6: WAMAL Training Setup

## A.3 AUXILEARN

AuxiLearn is a bi-level framework that uses implicit differentiation to learn new auxiliary tasks or weight existing tasks (Navon et al., 2020). Auxilearn introduced the use of auxiliary sets, which are small validation splits held out from the training data and used solely to tune the auxiliary-loss parameters. The Auxilearn researchers demonstrated that their procedure benefits from the use of an auxiliary set. Moreover, the researchers demonstrate that their framework surpasses MAXL in helpful auxiliary task creation.

### A.3.1 AUXILEARN EXPERIMENTS

In these experiments, we compare Auxilearn to WAMAL using their open-source Python optimizer. We tested their procedure with 10% auxiliary set against WAMAL.

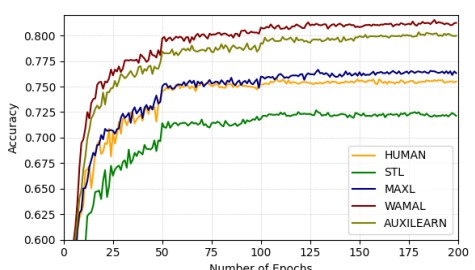
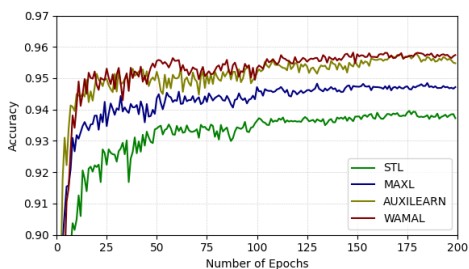

(a) CIFAR 100 20-Superclass - VGG16      (b) SVHN - VGG16

Figure 7: Sample Training Curves - Auxilearn

| Dataset | Method | Value |
|---|---|---|
| | WAMAL | 0.8153 |
| | AUXILEARN | 0.8033 |
| CIFAR100 20-Superclass | MAXL | 0.7665 |
| | HUMAN | 0.7577 |
| | STL | 0.7268 |
| | WAMAL | 0.9582 |
| | AUXILEARN | 0.9579 |
| SVHN | MAXL | 0.9485 |
| | STL | 0.9396 |

Table 3: Test Accuracies compared to Auxilearn

We see that our framework out performs Auxilearn from the results of the experiments given in Table 3 and Figure 7.

### A.3.2 AUXILIARY SET EXPERIMENTS

Auxilearn has been shown to benefit from the use of auxiliary sets. We verify the importance of the auxiliary set for Auxilearn on the 20-Superclass problem. Moreover, we test the impact of the use of a 10% auxiliary set on WAMAL. We also experimented with using auxiliary sets when fine-tuning ViT-B/16 on the Oxford-IIIT Pets and Food101 datasets.

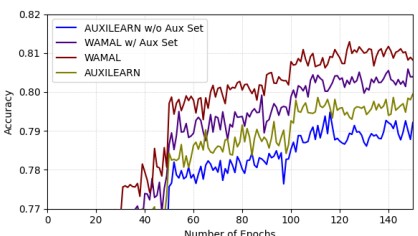
(a) CIFAR 100 20-Superclass - VGG16

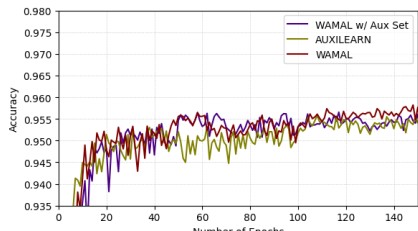
(b) SVHN - VGG16

Figure 8: Sample Training Curves - VGG16 w/ Auxiliary Set

| Method | SVHN | CIFAR100-20 |
|---|---|---|
| WAMAL | 0.9582 | 0.8153 |
| WAMAL w/ Aux Set | 0.9566 | 0.8074 |
| AUXILEARN | 0.9579 | 0.8033 |
| AUXILEARN w/o Aux Set | – | 0.7948 |

Table 4: Test accuracy of VGG16 using Auxiliary Sets

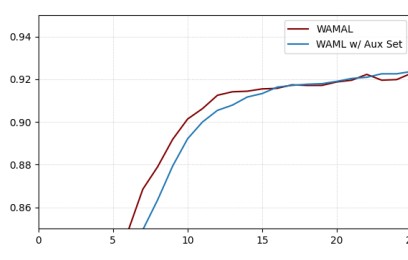
(a) Oxford-IIIT Pets - ViT-B/16

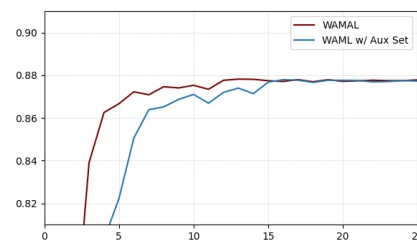
(b) Food101 - ViT-B/16

Figure 9: Sample Training Curves - ViT w/ Auxiliary Set

We find that WAMAL is not significantly positively impacted by the use of an Auxiliary Set.

| Dataset | Method | Value |
|---------|--------|-------|
| Oxford Pet | WAMAL | 0.9239 |
| | WAMAL w/ Aux Set | 0.9269 |
| Food101 | WAMAL | 0.8782 |
| | WAMAL w/ Aux Set | 0.8781 |

Table 5: Test Accuracies of ViT w/ Auxiliary Set

## A.4 STATISTICAL-SIGNIFICANCE

To confirm that the observed performance gaps are not due to random chance, we study the statistical significance of our findings. Across 3 runs for each of the tables below we collected the statistical indicators for our following experiments.

We list the sample mean, the sample standard deviation and the associated $95\%$ confidence interval obtained from the two-tailed Student $t$ distribution with $n - 1 = 2$ degrees of freedom. For every pair of methods we carry out a two-sided Welch $t$-test and report the mean difference $\Delta$, its $95\%$ confidence interval, the resulting $p$-value. All numerical results can be found in the tables below.

CIFAR-100 (20 SUPERCLASSES), VGG16 PRIMARY

Table 6: Test accuracy ($n$=3).

| Method | Accuracy |
|---|---|
| Single-Task | $0.7378 \pm 0.0028$ |
| Human labels | $0.7464 \pm 0.0017$ |
| MAXL | $0.7726 \pm 0.0005$ |
| **WAMAL** | **$0.8022 \pm 0.0023$** |

Table 7: Welch $t$-tests on CIFAR-100 accuracy.

| Comparison | $\Delta$ | 95% CI | $p$ |
|---|---|---|---|
| STL vs MAXL | $-0.0348$ | $[-0.0414, -0.0282]$ | $0.0016$ |
| STL vs WAMAL | $-0.0644$ | $[-0.0703, -0.0585]$ | $< 10^{-4}$ |
| MAXL vs WAMAL | $-0.0296$ | $[-0.0350, -0.0242]$ | $0.0013$ |

CIFAR-10, RESNET-50 PRIMARY

Table 8: Test accuracy ($n$=3).

| Method | Accuracy |
|---|---|
| Single-Task | $0.8783 \pm 0.0015$ |
| MAXL | $0.9035 \pm 0.0021$ |
| **WAMAL** | **$0.9114 \pm 0.0012$** |

Table 9: Welch $t$-tests on CIFAR-10 accuracy.

| Comparison | $\Delta$ | 95% CI | $p$ |
|---|---|---|---|
| STL vs MAXL | $-0.0252$ | $[-0.0295, -0.0210]$ | $1.2 \times 10^{-4}$ |
| STL vs WAMAL | $-0.0331$ | $[-0.0363, -0.0298]$ | $1.2 \times 10^{-5}$ |
| MAXL vs WAMAL | $-0.0078$ | $[-0.0120, -0.0036]$ | $0.009$ |

SVHN, VGG16 PRIMARY

Table 10: Test accuracy ($n=3$).

| Method | Accuracy |
|---|---|
| Single-Task | $0.9395 \pm 0.0003$ |
| MAXL | $0.9454 \pm 0.0012$ |
| **WAMAL** | **$0.9554 \pm 0.0012$** |

Table 11: Welch $t$-tests on SVHN accuracy.

| Comparison | $\Delta$ | 95% CI | $p$ |
|---|---|---|---|
| STL vs MAXL | $-0.0058$ | $[-0.0087, -0.0029]$ | $0.012$ |
| STL vs WAMAL | $-0.0158$ | $[-0.0185, -0.0132]$ | $0.0011$ |
| MAXL vs WAMAL | $-0.0100$ | $[-0.0127, -0.0073]$ | $5.2 \times 10^{-4}$ |

Across all three benchmarks WAMAL consistently achieves the highest mean accuracy. Every improvement over MAXL or single-task remains statistically significant, confirming the impact of our work.

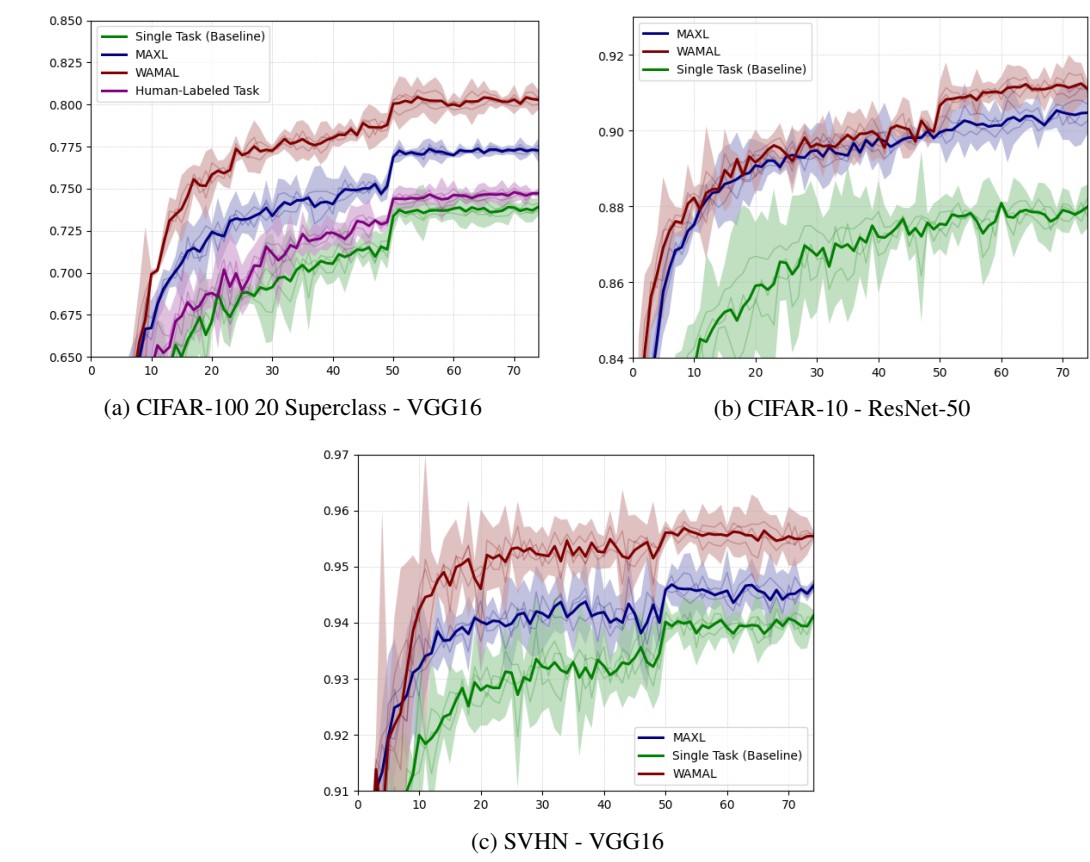

(a) CIFAR-100 20 Superclass - VGG16

(b) CIFAR-10 - ResNet-50

(c) SVHN - VGG16

Figure 10: Per-epoch test accuracy with 95% confidence intervals.

## A.5 WEIGHT DISTRIBUTIONS

The following are the bar graphs showing the distribution of weights as described in Section 6.2.

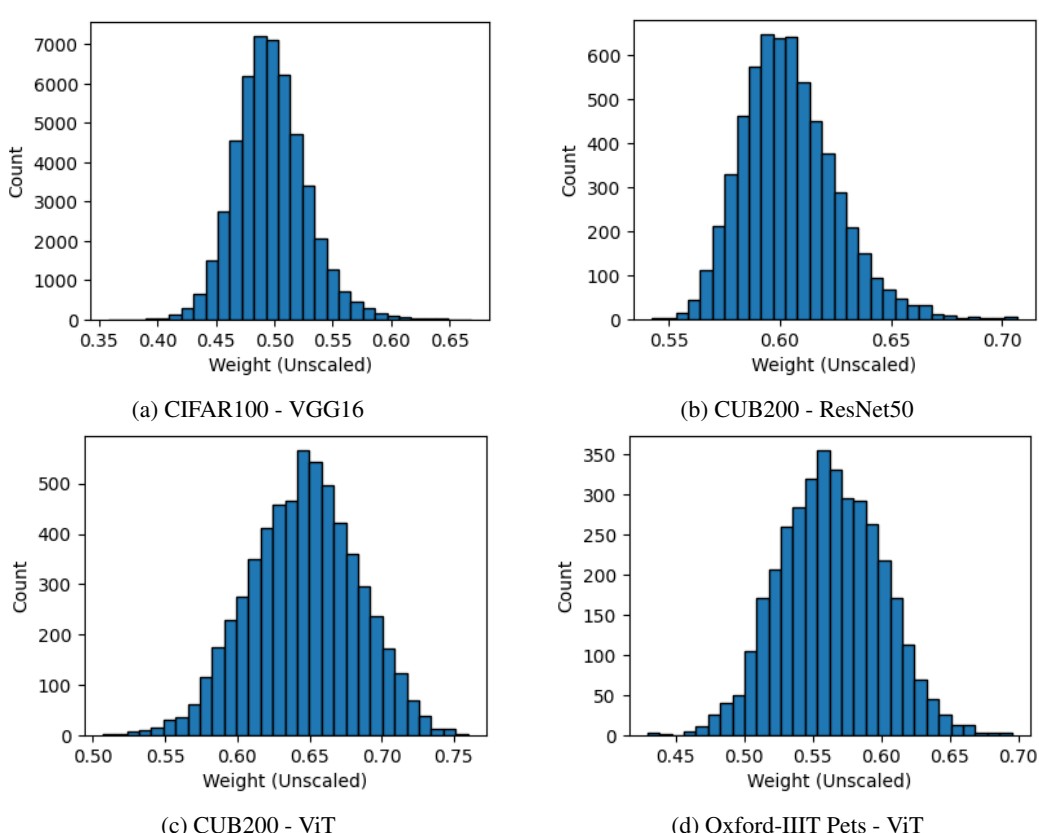

(a) CIFAR100 - VGG16

(b) CUB200 - ResNet50

(c) CUB200 - ViT

(d) Oxford-IIIT Pets - ViT

Figure 11: Weight Distributions from Label-Weight Network

## A.6 ADDITIONAL TEST ACCURACY CURVE SAMPLES

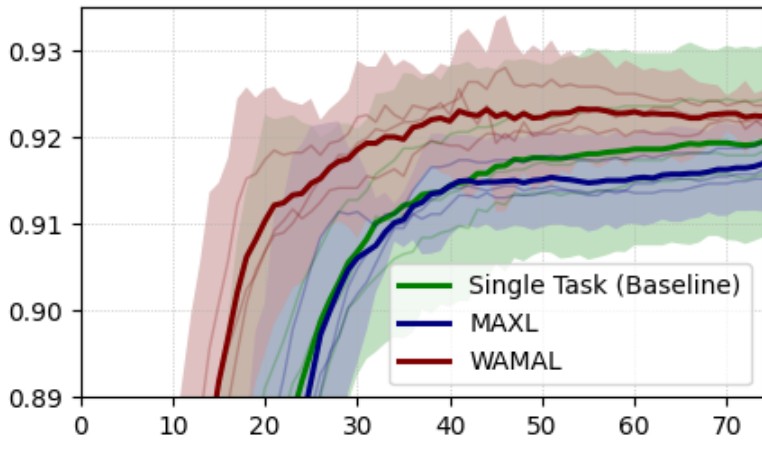

(a) Oxford-IIIT Pets (ViT-B/16, 30 epochs)

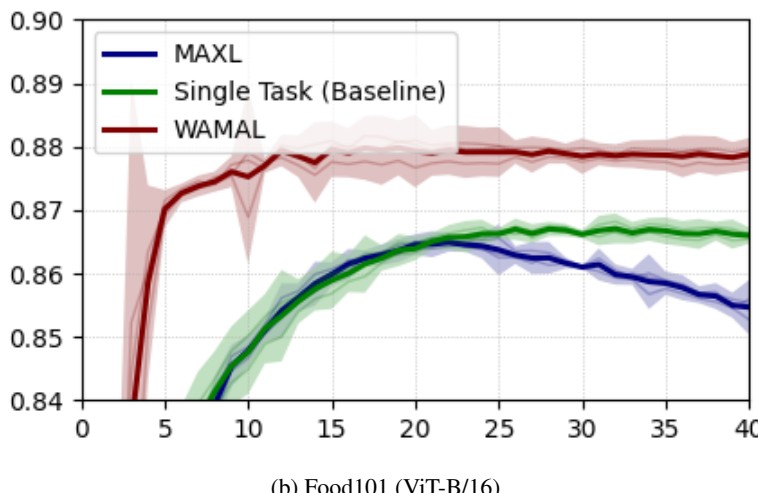

(b) Food101 (ViT-B/16)

Figure 12: Sample test accuracy curves for WAMAL, MAXL, and Single Task setups on Oxford-IIIT Pets (30 epochs) and Food101.

## A.7 ASYMMETRIC–BACKBONE ABLATION

In these experiments we study the impact of using primary and label network backbones of different types/sizes. For each primary network we replace the label/weight generator's backbone with a lighter or heavier architecture, keeping all other hyper-parameters fixed (Table 18) for the Cifar10 task.

Table 12: Effect of asymmetric backbones. Rows list the label and weight generating backbone. Columns list the primary backbone. Light generators (below the diagonal) provide the same benefit than heavier ones; oversizing yields little extra gain.

| Label-gen↓ | Primary backbone | | |
|---|---|---|---|
| | ResNet-18 | ResNet-50 | VGG-16 |
| ResNet-18 | 0.878 | 0.908 | **0.931** |
| VGG-16 | 0.881 | 0.914 | 0.929 |
| ResNet-50 | 0.883 | **0.917** | 0.930 |
| STL | 0.852 | 0.899 | 0.889 |

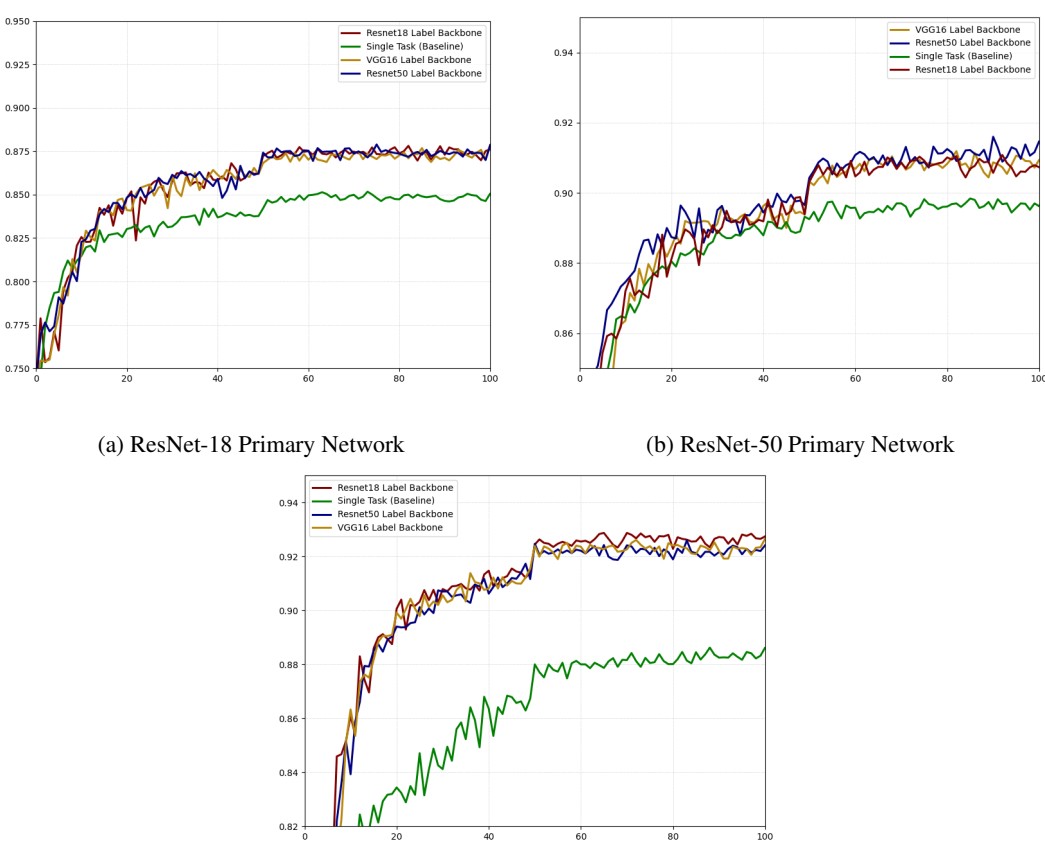

(a) ResNet-18 Primary Network        (b) ResNet-50 Primary Network

(c) VGG-16 Primary Network

Figure 13: Various primary network architectures with their accuracy while using different label network architectures.

We see that pairing a light ResNet-18 generator with any primary already yields strong gains. For example, ResNet-50 jumps from the STL baseline of 0.899 to 0.908. Meanwhile using larger generators offers, at best, marginal extra performance. Using a larger architecture for the label/weight generating network does not provide significant improvement to smaller networks. Every WAMAL configuration remains well above its single-task counterpart, so a small label/weight generator is clearly better than no WAMAL procedure. We leave a more exhaustive exploration into size trade-offs to future work.

## A.8 RANGE ABLATION

To understand how the weight bound $r$ influences stability and performance, we re-trained the VGG16 architecture on the CIFAR-100 20-Superclass task while sweeping $r \in \{0, 1.25, 2.5, 5, 10, 20, 40\}$. All other hyper-parameters were fixed to the defaults in Table 18. Our results can be found in Table 13 and in Figure 14.

Table 13: Impact of the log-range bound $r$ on CIFAR-100 20-Superclass (VGG16, last-15-epoch mean).

| Range $r$ | Accuracy (%) | Cross-entropy loss |
|---|---|---|
| 0 | 78.59 | 0.8148 |
| 1.25 | 79.78 | 0.7664 |
| 2.5 | 80.74 | 0.7025 |
| **5** | **81.60** | **0.6174** |
| 10 | 61.12 | 0.9257 |
| 20 | 73.24 | 0.5809 |
| 40 | 5.00 | 43.7491 |

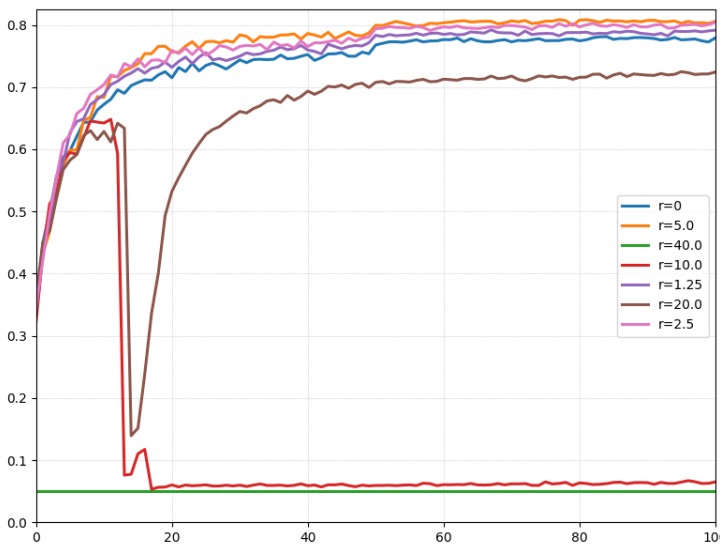

Figure 14: Range Ablation Accuracy Curves

We see that any dynamic re-weighting ($r > 0$) is better than a fixed auxiliary sample weight. Performance peaks at $r \approx 5$, aligning with our default. Beyond $r \geq 10$, the vast dynamic range magnifies the implicit degrades optimisation; at $r = 40$ training collapses entirely, empirically validating the stability bound of Section A.12.

### A.9 BATCH WEIGHT NORMALIZATION

In these experiments we examine the effect of normalizing the weights for the auxiliary task samples within a batch. We normalize such that the mean of the batch is equal to 1 after applying the weight scaling.

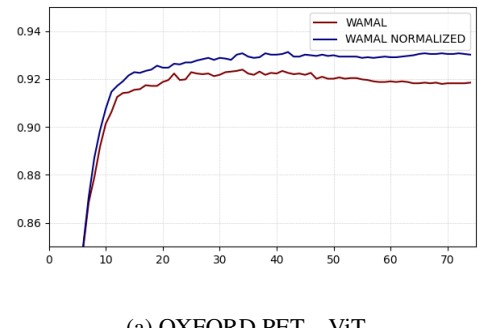

(a) OXFORD PET – ViT

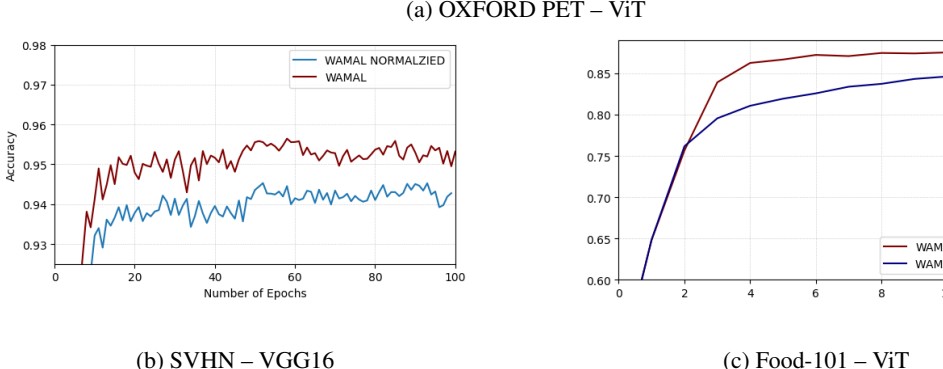

(b) SVHN – VGG16

(c) Food-101 – ViT

Figure 15: Sample Training Curves using Batch Weight Normalization.

Table 14: WAMAL with Batch Weight Normalization

| Method | SVHN | Food-101 | OXFORD PET |
|---|---|---|---|
| WAMAL | 0.9582 | 0.8782 | 0.9239 |
| WAMAL (Weight Normalised) | 0.9453 | 0.8583 | 0.9312 |

In Table 14 and Figure 15 we see that weight normalization is generally not beneficial. However, the Oxford-IIIT Pet dataset did benefit from normalization. This is likely likely due to the fact that the average of the learned weights for the Oxford-IIIT Pet dataset was close to 1, as we saw in the weight distribution exploration. Therefore, conversion of the weights to mean 1 did not have a negative impact on performance.

## A.10 ADAM OPTIMIZER

In these experiments we observe the impact of using Adam as the optimizer in place of SGD with WAMAL. We trained VGG16-based WAMAL architecture using a wide range of learning rates. Table 15 and Figure 16 demonstrate the findings of these experiments. We find that using Adam with WAMAL is feasible but requires careful selection of the Learning Rate, as the training setup is revealed to be extremely sensitive to changes in Learning Rate. Even with the extensive Learning Rate search, the results are still outperformed by SGD, which is more robust to varied Learning Rates.

With this knowledge, we attempted to fine-tune the ViT-B/16 using Adam with an optimized learning rate. Table 16 and Figure 17 present these results. We see that with careful selection of Learning Rate, fine-tuning of the ViT architecture yields strong results. The results on the Oxford-IIIT Pet dataset are even improved using Adam over SGD.

| Optimizer | Learning Rate | Test Accuracy |
|-----------|---------------|---------------|
| **SGD** | **0.01** | **0.8093** |
| ADAM | $2.5 \times 10^{-5}$ | 0.6773 |
| ADAM | $5 \times 10^{-5}$ | 0.7255 |
| ADAM | 0.00012 | 0.7742 |
| ADAM | 0.0003125 | 0.7930 |
| ADAM | 0.000625 | 0.7881 |
| ADAM | 0.00125 | 0.7681 |
| ADAM | 0.0025 | 0.6700 |
| ADAM | 0.005 | 0.6489 |
| ADAM | 0.01 | 0.5667 |
| ADAM | 0.02 | 0.5276 |
| ADAM | 0.04 | 0.3915 |

Table 15: Results of WAMAL Trained using Adam with various Learning Rates

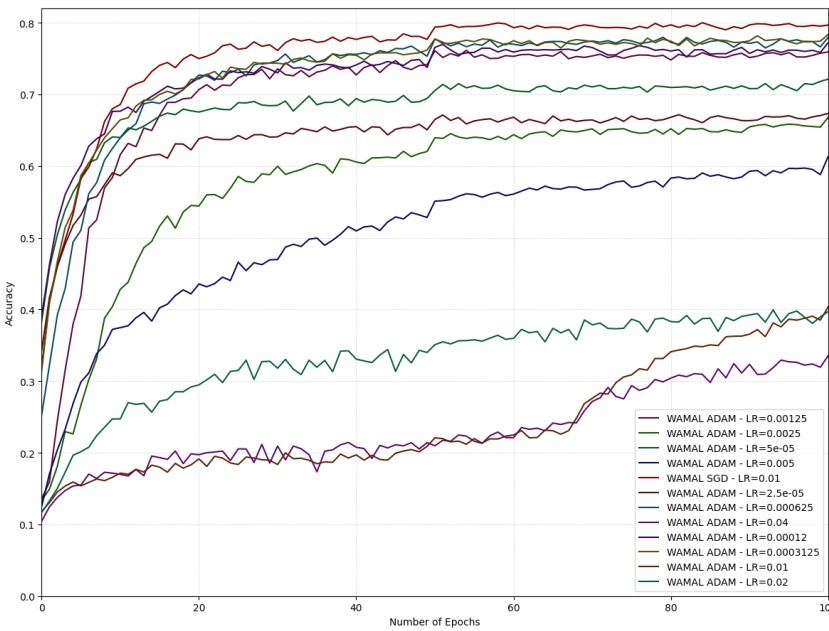

Figure 16: VGG16 CIFAR100 20-Superclass Test Accuracy - Adam Optimizer with Varied Learning Rates

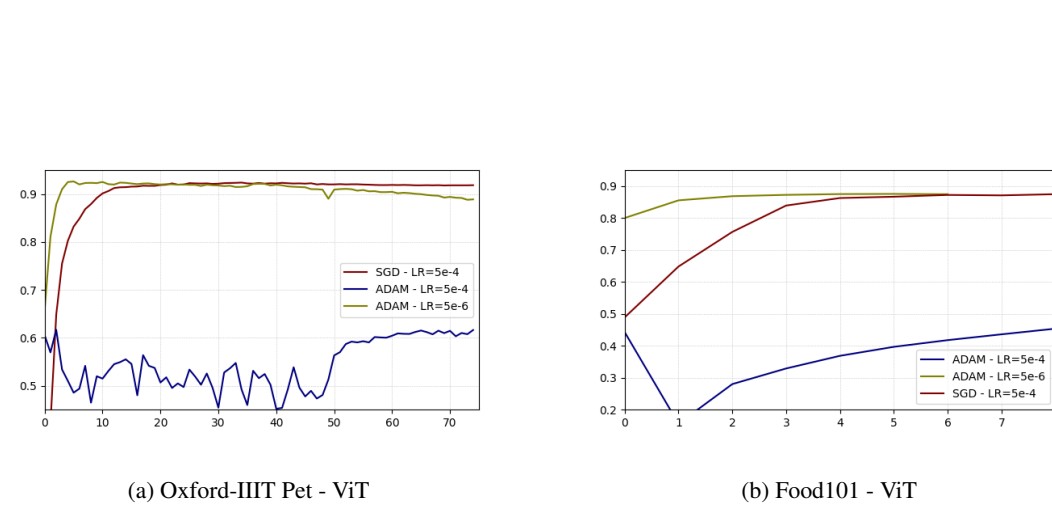

(a) Oxford-IIIT Pet - ViT                                    (b) Food101 - ViT

Figure 17: Sample Training Curves - ViT with Adam

| Dataset | Optimizer | Learning Rate | Value |
|---|---|---|---|
| **Food101** | **SGD** | $5 \times 10^{-4}$ | **0.8782** |
| Food101 | ADAM | $5 \times 10^{-6}$ | 0.8754 |
| Food101 | ADAM | $5 \times 10^{-4}$ | 0.4930 |
| Oxford Pet | SGD | $5 \times 10^{-4}$ | 0.9239 |
| **Oxford Pet** | **ADAM** | $5 \times 10^{-6}$ | **0.9264** |
| Oxford Pet | ADAM | $5 \times 10^{-4}$ | 0.6168 |

Table 16: Test accuracies fine-tuning ViT with Adam

## A.11 ADDITIONAL TRAINING DETAILS

Training was conducted on 2080Ti and 4090 GPU systems. Each experiment was run on a single GPU. The 4090's were used for the ViT-B/16 experiments and the 2080Tis were used for everything else.

Table 17: Per-experiment configuration summary. All runs used a single GPU; ViT-B/16 on 4090, others on 2080Ti.

| Task | Backbone | GPU | Img | Batch | Epochs |
|---|---|---|---|---|---|
| CIFAR100-20 | VGG16 | 2080Ti | 32 | 128 | 200 |
| SVHN | VGG16 | 2080Ti | 32 | 128 | 200 |
| CIFAR10 (ft) | ResNet50 | 2080Ti | 32 | 256 | 75 |
| CUB200 (ft) | ResNet50 | 2080Ti | 224 | 64 | 75 |
| Oxford-Pet (ft) | ViT-B/16 | 4090 | 224 | 64 | 75 |
| Food101 (ft) | ViT-B/16 | 4090 | 224 | 64 | 75 |

Table 18: Recommended default configuration (practical starting point).

| Component | Default |
|---|---|
| Optimizer | SGD |
| Primary LR | 0.01 |
| Label/Gen LR | $1 \times 10^{-3}$ |
| Label/Gen weight decay | $5 \times 10^{-4}$ |
| LR schedule | StepLR (step size 50, $\gamma = 0.5$) |
| Batch size | 100 |
| Hierarchy Factor | $\psi = 5$ |
| Entropy loss factor | 0.2 |
| Range $r$ | 5 |

Table 19: Notation summary.

| Symbol | Meaning |
|---|---|
| $D = \{(x_i, y_i)\}$ | dataset with inputs/primary labels |
| $Y, \hat{Y}$ | primary and auxiliary label sets |
| $f^\theta$ | primary network with primary/aux heads |
| $g^\phi$ | label/weight network with two heads |
| $\psi$ | subclasses per primary class (hierarchy factor) |
| $K$ | number of auxiliary classes ($K = \psi|Y|$) |
| $r$ | weight-range hyperparameter ($w \in [2^{-r}, 2^r]$) |

## A.12 BOUNDED-WEIGHT STABILITY PROPOSITION

**Assumptions.** We assume $\ell_{\text{prim}}$ and $\ell_{\text{aux}}$ are $\beta$-smooth in $\theta$, the masked softmax map is Lipschitz in its logits.

Let the total objective be

$$\mathcal{L}_{\text{total}}(\theta, \phi) = \mathcal{L}_{\text{prim}}(\theta, \phi) \; + \; w(\phi)\, \mathcal{L}_{\text{aux}}(\theta, \phi), \qquad w(\phi) \in [2^{-r}, 2^{\,r}],$$

and define the bilevel value function $g(\phi) = \mathcal{L}_{\text{aux}}\big(\theta^*(\phi), \phi\big)$ with $\theta^*(\phi) = \arg\min_\theta \mathcal{L}_{\text{total}}(\theta, \phi)$. Assume:

- $\mathcal{L}_{\text{prim}}(\theta, \phi)$ and $\mathcal{L}_{\text{aux}}(\theta, \phi)$ are $\beta$-smooth in $\theta$;
- $\big\|\nabla_\theta \mathcal{L}_{\text{aux}}(\theta, \phi)\big\| \le \gamma$;
- $\big\|\nabla_\phi \nabla_\theta \mathcal{L}_{\text{prim}}(\theta, \phi)\big\| \le \rho$;
- $\big\|\nabla_\phi \nabla_\theta \mathcal{L}_{\text{aux}}(\theta, \phi)\big\| \le \kappa$;
- $\big\|\nabla_\phi w(\phi)\big\| \le \omega$.

Then the gradient update

$$\nabla_\phi g(\phi) = -\nabla_\theta \mathcal{L}_{\text{aux}}\big(\theta^*(\phi), \phi\big) \big(\nabla_\theta^2 \mathcal{L}_{\text{total}}(\theta^*(\phi), \phi)\big)^{-1} \nabla_\phi \nabla_\theta \mathcal{L}_{\text{total}}\big(\theta^*(\phi), \phi\big)$$

is bounded

$$\big\|\nabla_\phi g(\phi)\big\| \;\le\; \frac{\gamma}{\beta}\Big(\rho \;+\; 2^{\,r}\kappa \;+\; \omega\,\gamma\Big).$$

*Proof.* By $\beta$-smoothness in $\theta$, $\big\|\nabla_\theta^2 \mathcal{L}_{\text{total}}(\theta^*(\phi), \phi)\big\| \le \beta$, so $\big\|\big(\nabla_\theta^2 \mathcal{L}_{\text{total}}(\theta^*(\phi), \phi)\big)^{-1}\big\| \le 1/\beta$. Expanding the mixed derivative gives

$$\nabla_\phi \nabla_\theta \mathcal{L}_{\text{total}}\big(\theta^*(\phi), \phi\big) = \nabla_\phi \nabla_\theta \mathcal{L}_{\text{prim}}\big(\theta^*(\phi), \phi\big) + w(\phi)\, \nabla_\phi \nabla_\theta \mathcal{L}_{\text{aux}}\big(\theta^*(\phi), \phi\big) + \big(\nabla_\phi w(\phi)\big)\big(\nabla_\theta \mathcal{L}_{\text{aux}}\big(\theta^*(\phi), \phi\big)\big)^\top.$$

Taking norms and using the assumptions with $w(\phi) \le 2^{\,r}$ yields

$$\big\|\nabla_\phi \nabla_\theta \mathcal{L}_{\text{total}}\big(\theta^*(\phi), \phi\big)\big\| \le \rho + 2^{\,r}\kappa + \omega\,\gamma.$$

Therefore,

$$\big\|\nabla_\phi g(\phi)\big\| \le \big\|\nabla_\theta \mathcal{L}_{\text{aux}}(\theta^*(\phi), \phi)\big\| \, \big\|\big(\nabla_\theta^2 \mathcal{L}_{\text{total}}(\theta^*(\phi), \phi)\big)^{-1}\big\| \, \big\|\nabla_\phi \nabla_\theta \mathcal{L}_{\text{total}}\big(\theta^*(\phi), \phi\big)\big\| \le \frac{\gamma}{\beta}\big(\rho + 2^{\,r}\kappa + \omega\,\gamma\big). \quad \square$$

This tells us that $r$ provides a bound on the gradient update with respect to the auxiliary task labels/network (via the $2^{\,r}$ factor).

## A.13 WAMAL ALGORITHM

---

**Algorithm 1** WAMAL (outer: label/weight net $g^\phi$, inner: primary $f^\theta$)

---

1: **Input:** Dataset $D$, hierarchy $\psi$, weight range $r$, losses $\ell_{\text{prim}}, \ell_{\text{aux}}$
2: Initialize $\theta, \phi$
3: Split $D$ into $D_{meta}$ and $D_{prim}$
4: **repeat**
5:    **(Inner step: update $\theta$ for primary)**
6:    **repeat**
7:       Sample minibatch $B_{prim} \subset D_{prim}$
8:       Compute per-sample loss $\ell_{\text{prim}}(f^\theta(x), y) + g^\phi_{\text{weight}}(x)\, \ell_{\text{aux}}(f^\theta(x), g^\phi_{\text{label}}(x))$
9:       Update $\theta \leftarrow \theta - \eta_\theta \nabla_\theta \ell_{\text{total}}$
10:   **until** All $D_{prim}$ batches are used
11:   **(Outer step: update $\phi$ for labels/weights)**
12:   **repeat**
13:       Sample minibatch $B_{meta} \subset D_{meta}$
14:       Compute $g^\phi_{\text{label}}(x, y; \psi)$, $g^\phi_{\text{weight}}(x)$ for $x \in B_{meta}$
15:       Compute $\widehat{\nabla_\phi}$ via Eq. equation 4
16:       Update $\phi \leftarrow \phi - \eta_\phi \widehat{\nabla_\phi}$
17:   **until** All $D_{meta}$ batches are used
18: **until** epochs complete

---

## B    ETHICS STATEMENT

We have no additional ethics concerns beyond the ones presented in the main body of the paper. We believe we are adhering to the Code of Ethics of ICLR.

## C    REPRODUCIBILITY STATEMENT

We provided full reproducibility for our work. We provided an anonymized version of our repo that can be used to recreate our experiments. We provide full information around our hyperparameter selection and the nature of the data used.

