# OpenReview forum: "Weight-Aware Meta Auxiliary Learning"
_ICLR.cc/2026/Conference — Submitted to ICLR 2026_

### Official Review · Reviewer_BxVj · 2025-10-15

**Soundness:** 2
**Presentation:** 1
**Contribution:** 1
**Rating:** 2
**Confidence:** 4

**Summary:**

The paper considers Auxiliary Learning (AL), a method for improving a model’s primary task by using additional auxiliary tasks for extra supervision. It proposes weight-aware meta auxiliary learning (WAMAL), which jointly learns auxiliary labels and per-sample weights through bi-level optimization. The paper focuses on image classification tasks and provides experiments on several datasets, which show improved performance compared to baselines.

**Strengths:**

1. The paper is easy to read.
1. WAMAL introduces the first approach that explicitly learns both auxiliary labels and per-sample weights.
1. Experiments show the method outperforms the single-task and MAXL baselines.

**Weaknesses:**

This paper presents several notable weaknesses in terms of novelty, experimental depth, and presentation quality. Overall, the work feels too preliminary.

1. My concern is the paper’s novelty: The conceptual and technical novelty beyond prior work, particularly MAXL (Liu et al., 2019) and AuxiLearn (Navon et al., 2020), appears modest. The proposed WAMAL framework primarily extends MAXL by introducing learned per-sample auxiliary weights, but the underlying optimization paradigm remains largely unchanged. Furthermore, AuxiLearn has already shown that learning task weights is equivalent to per-datum adaptive weighting, and experimented with learning both auxiliary tasks and their respective weights.
1. Experimental evaluation is limited for a pure empirical paper:
    - Task diversity: All experiments are confined to image classification. The work would benefit from evaluations on additional modalities (e.g., text) or tasks (e.g., semantic segmentation, depth estimation etc.).
    - Baselines:  In the main text, WAMAL is compared only to single-task learning and MAXL, i.e., a single auxiliary-learning baseline. This is insufficient for a fair empirical study. While the appendix briefly includes AuxiLearn, comprehensive comparisons should appear in the main text and across all datasets. Please add additional AL baselines to the main text.
1. The related work section omits several important and conceptually relevant works:
     - Auxiliary Learning with Joint Task and Data Scheduling, ICML 2022.
     - Auxiliary Learning as an Asymmetric Bargaining Game, ICML 2023.
     - Meta-Weight-Net: Learning an Explicit Mapping For Sample Weighting, NeurIPS 2019. (for learned per-sample weighting)
     - And some more classic AL papers like: Adapting Auxiliary Losses Using Gradient Similarity (2018), Adaptive auxiliary task weighting for reinforcement learning (NeurIPS 2019), etc.

**Questions:**

Other Concerns:
1. Paper structure: Some implementation details (for example, the section on the Python wrapper) are better suited for the appendix, freeing space in the main text for more substantial empirical results and discussion.
1. Presentation: Important experimental results appear only in the appendix (with relevant text in the main paper) and should be moved to the main paper for clarity and impact.
1. Computational cost: The proposed framework introduces computational overhead (additional network and bi-level optimization). The paper does not report or analyze this additional cost, which may limit the method’s practicality and scalability.

**Details Of Ethics Concerns:**

Authors discuss ethics-related concerns.

---

### Official Review · Reviewer_eejB · 2025-10-29

**Soundness:** 2
**Presentation:** 2
**Contribution:** 2
**Rating:** 2
**Confidence:** 3

**Summary:**

This paper introduces Weight-Aware Meta Auxiliary Learning (WAMAL), a bi-level optimization framework that simultaneously generates auxiliary labels and per-sample weights for auxiliary learning. The method is evaluated on standard image classification datasets including CIFAR-100-20, SVHN, Oxford-IIIT Pet, Food-101, and CUB-200 using VGG16, ResNet50, and ViT-B/16 architectures. The authors report consistent accuracy improvements over single-task training and meta-auxiliary baselines. An open-source wrapper is provided to convert existing image classifiers into WAMAL-ready networks.

**Strengths:**

1. The paper is clearly written and easy to follow.
2. Consistent but small accuracy gains on CIFAR-100-20, SVHN, Pets, Food-101 when trained from scratch or fine-tuned.

**Weaknesses:**

1. Motivation. Auxiliary learning method is designed to decide which label to use (mentioned in motivation 1), I don't this method can interpret the question better than previous auxiliary methods.
2. The designed method is simple, thus it lacks novelty.
3. In the experiment part, the authors only choose two baselines to be compared, which makes the result not convincing.
4. In Figure 3, the authors paint high/low weight picture in different datasets, but it is hard to tell why the top ones should have low weight while the bottom ones should have high weights.

**Questions:**

See weakness

---

### Official Review · Reviewer_qd7C · 2025-10-30

**Soundness:** 3
**Presentation:** 3
**Contribution:** 2
**Rating:** 6
**Confidence:** 2

**Summary:**

This paper proposes weight-aware meta auxiliary learning, which simultaneously learns auxiliary labels and the auxiliary loss weight for each sample to enhance the performance of the main task. This method addresses the reliance of existing auxiliary learning approaches on manually annotated auxiliary tasks and the uniform weighting of auxiliary losses across all samples. By enabling more fine-grained and adaptive supervision, the proposed approach can effectively guide the main task and achieve improvements over previous methods.

**Strengths:**

- The proposed two-level optimization framework and sample-adaptive weighting can effectively enhance the performance of the method, achieving relatively good results.
- The paper is relatively well-organized and easy for subsequent work to follow.

**Weaknesses:**

- The abstract devotes a large portion to describing the background and issues of existing methods. However, it provides insufficient description of the innovations and contributions of this work, making it less compelling.
- Figure 1 appears somewhat confusing; it does not clearly convey the interaction between the weights of the label network and the main network. How does it function? The current single arrow without any explanation is unclear. In addition, some extra explanation should be added to aid understanding of the label generation process.
- It is not surprising that adding an extra weight from the samples to the network would improve performance. The authors experimentally demonstrate the effectiveness of distinguishing the number of categories through parameter settings, but is there any theoretical support for this? The authors should clarify.
- In the auxiliary weight supervision, how is the weight obtained? How is it ensured that this weight accurately reflects the importance of the sample?

**Questions:**

Please refer to the weaknesses section.

---

### Official Review · Reviewer_xXBi · 2025-10-31

**Soundness:** 2
**Presentation:** 3
**Contribution:** 2
**Rating:** 4
**Confidence:** 4

**Summary:**

Summary

This paper falls within the area of Auxiliary Learning, a subfield of Multi-Task Learning (MTL) aimed at improving the generalization of a main task by leveraging auxiliary tasks.
The authors propose Weight-Aware Meta Auxiliary Learning (WAMAL) to address two challenges:

Mitigating negative interference from auxiliary tasks.

Handling the lack of labeled data for auxiliary tasks.

The approach combines sample-level weighting, which is known to be more robust to noise than task-level weighting, with the Meta Auxiliary Learning (MAXL) algorithm.
Experiments compare WAMAL against MAXL and single-task learning baselines on two datasets, including fine-tuning analyses.

**Strengths:**

Strengths

The idea of combining sample-level weighting and auxiliary-task label generation is novel.

The reported results are favorable for WAMAL, showing improved generalization over single-task learning and MAXL.

The authors include some ablation studies that analyze hyperparameter ranges and optimizer choice.

**Weaknesses:**

Weaknesses

The connection to prior weighting algorithms is underdeveloped. MTL methods often use gradient similarities to guide task weighting; this relationship should be clearly discussed.

The rationale for sample-level weighting is unclear. The design motivation and computation of these weights need to be explicitly justified.

Benchmarking is limited, with comparisons to only MAXL. Including additional auxiliary-learning baselines (both in datasets and weighting schemes) would make the evaluation more convincing.

The structure of the paper is confusing. Related work and methodology are interwoven, making it hard to identify what is new relative to MAXL.

Writing and notation require refinement. For example, the paper switches between upper- and lowercase L  (possibly total vs. sample-level weights) without clear definition or aggregation details.


While the method’s motivation is reasonable, the design decisions are weakly justified and the contribution is largely compositional.
The paper does not convincingly explain why these specific components (sample-level weighting and MAXL) belong together or how they interact.
Given the modest novelty, limited evaluation, and writing issues, the contribution currently lacks the rigor needed for acceptance.

**Questions:**

Questions for the Authors

What is the reasoning behind the sample-level weighting method? How does it relate to other sample-level weighting approaches in single- or multi-task learning?

Is it surprising that the model learns an approximately normal distribution of weights? Would randomly sampled weights yield similar results, as sometimes observed in MTL?

Can you quantitatively verify that lower weights correspond to noisier samples?

What are the exact differences between MAXL and WAMAL beyond the second head predicting sample-level weights? Please clarify this in Section 3.1.

What is the theoretical motivation for Equation (4)?

How do bounded weights prevent dominance or vanishing of the auxiliary term? Can this be shown empirically?

Could the framework easily incorporate alternative weight generation mechanisms?

Additional Feedback

The writing requires improvement: the abstract and literature review are too limited and occasionally repetitive. The tone could be more formal.

The structure should be revised to clearly separate related work from methodology.

The repository link was not functional.

The paper should explicitly list its key contributions and clarify which components are new.

The literature review lacks nuance: sample-level weighting is not a standard practice in auxiliary learning and should not be presented as such.

---

### Meta-Review · Area_Chair_oMrB · 2025-12-04

**Summary:**

The submission received majorly negative ratings. The recommended rejection is informed by the convergence of reviewer concerns across three core drawbacks: limited novelty, insufficient experimental validation, and presentational weaknesses. All reviewers, despite varying scores, agreed that the core technical advancement that integrating per-sample weighting into a meta-auxiliary learning (MAXL) framework is incremental. Its novelty and necessity over prior art (e.g., MAXL, AuxiLearn) are not convincingly established. The empirical evaluation was broadly criticized as narrow, being confined to image classification and lacking comparisons against a comprehensive suite of modern baselines from auxiliary and multi-task learning. Furthermore, reviewers highlighted significant issues with the paper's structure, clarity of motivation, and justification of design choices, which hindered a clear assessment of its contribution.

**Reviewer Concerns:**

The authors did not provide a full rebuttal. They only provided a response about "Auxiliary Learning Objective". It primarily addressed concerns about the selection of baselines and the statistical significance of results, arguing that their chosen comparisons (STL, MAXL, AuxiLearn) are the most directly relevant in the meta-auxiliary learning sub-field and that their performance gains are meaningful. However, they did not adequately resolve the more fundamental, outstanding concerns. It did not strengthen the justification for the novel composition of sample-level weighting with label generation, nor did it expand the limited experimental scope to other modalities or tasks. Critiques regarding the underdeveloped connection to related weighting literature (e.g., gradient similarity methods, Meta-Weight-Net), the theoretical motivation for key design equations, and the overall presentation and structure of the paper remain largely unaddressed.

**Reviewer Scores:**

For Reviewer xXBi (4), Reviewer qd7C (6), Reviewer BxVj (2), they probably will not change their mind, as the authors did not provide responses in the discussion period.

Reviewer eejB (2) might increase his/her rating (say to 4) due to the response from authors. But I am not sure if it will happen due to other ratings.

---

### Decision · Program_Chairs · 2026-01-26

Reject